# Hybrid Hydrogels for Neomycin Delivery: Synergistic Effects of Natural/Synthetic Polymers and Proteins

**DOI:** 10.3390/polym15030630

**Published:** 2023-01-26

**Authors:** Maria Bercea, Ioana-Alexandra Plugariu, Luiza Madalina Gradinaru, Mihaela Avadanei, Florica Doroftei, Vasile Robert Gradinaru

**Affiliations:** 1“Petru Poni” Institute of Macromolecular Chemistry, 41-A Grigore Ghica Voda Alley, 700487 Iasi, Romania; 2Faculty of Chemistry, Alexandru Ioan Cuza University of Iasi, 11 Carol I Bd., 700506 Iasi, Romania

**Keywords:** hybrid hydrogel, PVA, PULL, BSA, lysozyme, viscoelasticity, swelling, neomycin release

## Abstract

This paper reports new physical hydrogels obtained by the freezing/thawing method. They include pullulan (PULL) and poly(vinyl alcohol) (PVA) as polymers, bovine serum albumin (BSA) as protein, and a tripeptide, reduced glutathione (GSH). In addition, a sample containing PULL/PVA and lysozyme was obtained in similar conditions. SEM analysis evidenced the formation of networks with porous structure. The average pore size was found to be between 15.7 μm and 24.5 μm. All samples exhibited viscoelastic behavior typical to networks, the hydrogel strength being influenced by the protein content. Infrared spectroscopy analysis revealed the presence of intermolecular hydrogen bonds and hydrophobic interactions (more pronounced for BSA content between 30% and 70%). The swelling kinetics investigated in buffer solution (pH = 7.4) at 37 °C evidenced a quasi-Fickian diffusion for all samples. The hydrogels were loaded with neomycin trisulfate salt hydrate (taken as a model drug), and the optimum formulations (samples containing 10–30% BSA or 2% lysozyme) proved a sustained drug release over 480 min in simulated physiological conditions. The experimental data were analyzed using different kinetic models in order to investigate the drug release mechanism. Among them, the semi-empirical Korsmeyer–Peppas and Peppas–Sahlin models were suitable to describe in vitro drug release mechanism of neomycin sulfate from the investigated hybrid hydrogels. The structural, viscoelastic, and swelling properties of PULL/PVA/protein hybrid hydrogels are influenced by their composition and preparation conditions, and they represent important factors for in vitro drug release behavior.

## 1. Introduction

The use of polysaccharides and synthetic polymers in hydrogels is advantageous for the design of biomaterials due to synergetic combination of their specific properties: biocompatibility, biodegradability, and non-toxicity of natural polymers and mechanical strength induced by synthetic polymers [1,2,3,4,5]. In recent decades, versatile hydrogels with elasticity close to that of natural tissues were obtained by physical, chemical, or combined strategies. Due to their hydrophilic and porous structure, the hydrogels can incorporate a large amount of water or physiological fluids and thus, they are suitable for various biomedical applications: systems for drug delivery, wound healing, regenerative medicine, biosensors, artificial contact lenses etc.

Wound healing is a dynamic and complex process of restoring skin cellular structures. An appropriate wound dressing maintains a wet environment at the wound surface, presents gas permeability, acts as a barrier to bacteria, and removes excess exudates [5,6,7,8]. A variety of strategies are applied to treat wound infections, using either biomaterials with inherent antimicrobial activity, or matrices for loading and releasing the drugs or other antimicrobial agents [9].

Due to its ability to form gels by physical interactions, poly(vinyl alcohol) (PVA) was previously tested for wound dressing applications [10]. The networks can be obtained by physical, chemical, or combined methods. Physical methods avoid the potential toxicity of chemical routes of synthesis, and the properties of hydrogels fit the requirements of targeted applications. PVA solutions can generate stable hydrogels when subjected to a certain number of freezing/thawing cycles followed by aging at a given temperature (usually room temperature or 37 °C). The microcrystalline junctions formed during this process act as coupling points in the physical network. Polymer concentration, the number of freezing/thawing cycles and the aging temperature determine the morphology and stability of hydrogels [11,12,13,14,15,16,17,18].

However, single PVA-based networks provided unsatisfactory swelling and biological properties as wound dressing systems. To overcome this deficiency, PVA was combined with other water-soluble natural and synthetic polymers, including sodium alginate, sodium carboxymethyl cellulose, dextran, chitosan, poly(vinylpyrrolidone), Pluronics, and polyurethanes, to prepare wound dressing systems [5,6,10,12,15,16,17,18,19]. For a series of hydrogels containing polymer mixtures, the pharmaceutical, swelling, and mechanical properties were enhanced to a lesser degree than expected [10]. The addition of biomolecules to PVA hydrogels represents a suitable alternative to fulfill the dressing requirements: high water content, softness, flexibility, and biocompatibility [3,8,15,19,20]. Pullulan and its derivatives are included in a variety of biomaterials with a high potential for wound dressing applications [21]. The presence of bulky pullulan macromolecules in the hybrid materials prevents the molecular packing around proteins, making this polysaccharide a suitable pharmaceutical excipient for protein stabilization [22].

Proteins and peptides have a therapeutic effect; they improve tissue repair and present a high potential for wound healing [23,24]. Bovine serum albumin (BSA) is a valuable component for biomedical and pharmaceutical formulations due to its biological properties, water solubility, and easy incorporation into materials [19,24,25]. Investigations of BSA conformation in the presence of different drugs (amoxicillin, cephalexin, and azithromycin) by circular dichroism spectroscopy coupled with conductive atomic force microscopy have shown that the protein–drug interactions alter the secondary structure of BSA due to the structural transitions; the α-helix and β-sheet contents were changed by up to 6% after the drug binding [26]. A tripeptide of interest is L-γ-glutamyl-L-cysteinyl-glycine (reduced glutathione, GSH), which has an important role in several maladies (including neurodegeneration and cancer), acting as reducing/antioxidant agent inside the cells. It also plays a mediator role in other physiological processes, such as cellular signaling, thiol-disulfide exchanges, and metabolism of xenobiotics, being an important reservoir of cysteine [27]. In biological systems, low GSH content has regulatory effects, improving tissue regeneration, wound healing, or dilation of blood vessels, whereas a high GSH content induces antimicrobial activity [28,29]. The GSH molecules are involved in intermolecular crosslinking of proteins in the hybrid polymer/protein hydrogels, and they improve the therapeutic effect of the drug-loaded material [29], induce the self-healing ability of the polymer/protein network [25,30], and regulate the formation of disulfide bonds with proteins [31].

Lysozyme is a bacteriolytic enzyme that is found in large quantities in secretions (tears, saliva and mucus, in the cytoplasm of polymorphonuclear neutrophils etc.) or in appreciable amount in chicken eggs. It has the ability to destroy the cell wall of Gram-positive bacteria by hydrolyzing the β-1,4 glycosidic bond between N-acetylmuramic acid and N-acetylglucosamine (carbohydrates present in the bacterial wall). Hybrid polymer/protein hydrogels with tunable dynamics can promote wound healing through their intrinsic characteristics or by releasing the entrapped drugs and biomolecules [7].

Therapeutic agents, such as antibiotics, are frequently incorporated into the dressings to prevent and treat wound infections [32,33]. As an example, electrospun PVA/sodium alginate mats loaded with drugs act as efficient barrier to the wound region during the formation of new tissue [15]. A promising formulation for buccal mucosal wound healing was obtained by loading neomycin sulfate (denoted as neomycin) into a gel with solid lipid nanoparticles [34]. Thus, neomycin proved to be an efficient drug in hydrogel dressings [10] or when it was used in combination with silver nanoparticles [35] for wound management. Neomycin was clinically tested and designed as useful drug against infections due to *Trypanosoma cruzi* [36] or certain Gram-positive bacteria (such as *Staphylococcus aureus* [37]), but it was found to be more efficient against most Gram-negative organisms (*Proteus vulgaris*, *Escherichia coli*, *Aerobacter aerogenes, P. aeruginosa* etc.), without toxic effects [38,39,40,41]. In aquaculture, neomycin diminished the oxidative damage and shown non-specific immunity in bacterial and viral infections in fish (*Carassius auratus gibelio*) [42]. Veterinarians recommend the administration of neomycin to treat localized skin infections and bacterial infections in the intestines of cats and dogs [43].

In the present paper, novel neomycin-loaded systems are reported. They consist of PVA/PULL/protein/peptide hydrogels, using BSA or lysozyme as proteins and GSH as peptide. The release of neomycin sulfate was investigated in simulated physiological conditions. Thus, the specific objective of this work was to design porous hybrid hydrogels and to study the neomycin delivery from these materials.

## 2. Materials and Methods

### 2.1. Materials

Poly(vinyl alcohol) (PVA, 130 kg/mol, 99% hydrolyzed), bovine serum albumin (BSA, 66.3 kg/mol, 98% purity), and reduced glutathione (GSH, 307 g/mol) were purchased from Sigma-Aldrich (Taufkirchen, Germany) and used as received. Lysozyme was purchased from Carl Roth (Karlsruhe, Germany). A pullulan (PULL) sample of 300 kg/mol was purchased from TCI Europe N.V. (Zwijndrecht, Belgium). Sodium phosphate buffer solution (PBS), pH = 7.4, was prepared by following the standard protocol, with dihydrate monosodium phosphate (NaH_2_PO_4_ × 2H_2_O) and pentahydrate disodium phosphate (Na_2_HPO_4_ × 7H_2_O) dissolved in Millipore water.

Neomycin trisulfate salt hydrate (denoted as neomycin for short) was purchased from Sigma-Aldrich (Taufkirchen, Germany). It is an aminoglycoside antibiotic, soluble in water (50 mg/mL), that enhances the cationic lipid-mediated transfection efficiency of reporter plasmids and oligonucleotides [44].

The structures of the chemicals used in this study are presented in Figure 1.

First, 5% wt. homogeneous solutions of PVA, PULL and BSA were freshly prepared in PBS. PVA solution was prepared at 90 °C by using magnetic stirring until the polymer was completely dissolved; then, the solution was cooled down to room temperature and left to equilibrate until the next day. BSA, lysozyme and PULL were dissolved in PBS at room temperature by gentle shaking with a rolling mixer, then the homogeneous solutions were stored for 24 h in refrigerator at 4 °C. All solutions were filtered by using a 0.45 μm Millipore filter. A stock solution of 5% wt. polymer concentration, containing 25% PULL and 75% PVA (wt./wt.), was further used for the preparation of hybrid hydrogels. The corresponding amounts of PULL/PVA solution were mixed with 5% wt. BSA solution, and samples with different concentrations of BSA (*w*_BSA_, % wt.) were prepared (Table 1). In addition, 1 mmol/L GSH (content found in healthy human cells) was added to all samples that were further stirred for 12 h at room temperature.

### 2.2. Hydrogel Preparation

The mixtures formed by homogeneous polymer solutions (PVA and PULL), proteins (BSA or lysozyme) and tripeptide (GSH) were submitted to 3 successive freezing/thawing cycles that allow for the formation of physical networks. The samples were suddenly frozen in liquid nitrogen and then gradually thawed to 37 °C for 24 h.

The freeze-drying of the hydrogel samples was carried out at −57 °C and 5.5 × 10^−4^ mbar by using ALPHA 1–2 LD Christ lyophilizer (Osterode, Germany).

### 2.3. Scanning Electron Microscopy Studies

The morphology of hydrogels was examined on cross-sections of freeze-dried samples by using Verios G4 UC Scanning Electron Microscope (Thermo Scientific, Brno, Czech Republic), operating at 5 kV in High Vacuum mode with secondary electrons detector (Everhart–Thornley detector, ETD). The SEM images were analyzed at various magnifications. Image J software was used to determine the average pore sizes from the SEM micrographs, and each value was obtained as an average of at least 80 dimensions.

### 2.4. Fourier Transform Infrared Spectroscopy

The infrared spectra of the hydrogel samples were obtained in Attenuated Total Reflectance (ATR) geometry by using a Bruker Vertex 70 spectrometer (Germany) equipped with a Specac™ ATR unit. The sample surface was scanned, with a resolution of 2 cm^−1^ in the 4000 cm^−1^–600 cm^−1^ range of wavenumber. The peaks of interest were resolved by Fourier self-deconvolution and curve-fitting with Voigt function, where the number and position of the component bands were taken from the second derivative of the spectra. The fractional area of the curve-fitted component was used to determine the percentage of the secondary structure elements of BSA. The difference spectra [hydrogel + BSA]—[control] and [hydrogel + BSA]—[BSA] were calculated using the wagging vibration of the combined C-O + C-C bonds at 844 cm^−1^ as internal standard. All operations were performed with OPUS 6.6 software from Bruker GmBH.

### 2.5. Rheological Investigation

The rheological measurements were carried out by using a MCR 302 Anton Paar rheometer (Graz, Austria), equipped with plane–plane geometry (an upper plate diameter of 50 mm, gap of 500 μm) and a Peltier device for temperature control. In order to limit the experimental errors, an anti-evaporation device was used to create a saturated atmosphere of solvent in the neighborhood of the sample. Prior to samples formulation, a preliminary rheological study was conducted for various PVA/PULL mixtures as solutions or physical networks in order to select the most suitable composition for hydrogel preparation. The results will be published elsewhere.

The rheological behavior of hydrogels was studied at 37 °C in oscillatory shear conditions. The linear range of viscoelasticity was determined using preliminary amplitude sweep tests. The viscoelastic moduli, G′ and G″, and the loss tangent (tan*δ*) were determined in frequency sweep experiments for oscillation frequencies (ω) varying from 0.1 rad/s to 100 rad/s (at constant deformation, γ, of 1%). G’ is the elastic modulus; it is a measure of the stored energy over one cycle of deformation. G″ is the viscous modulus, and gives indication of the dissipated energy over one cycle of deformation. The G″/G′ ratio is the loss tangent and states the degree of viscoelasticity of the sample.

The shear viscosity (η) was measured in stationary flow conditions for different shear rate ($\dot{\gamma }$) values ranging from 0.01 s^−1^ to 100 s^−1^.

### 2.6. Swelling Behavior

The swelling of hydrogels was studied at 37 °C in PBS (pH = 7.4) for samples that were previously freeze-dried. After checking the exact weight of the dry sample (*m_o_*), it was immersed in buffer solution. The hydrogel was taken out at different time intervals and its weight was determined (*m_t_*). The tests were repeated three times for each sample.

The swelling degree (*S*) is defined as the ratio between the solvent weight in the swollen sample at a given time *t* (*m_t_*) and the weight of the corresponding dried sample (*m_o_*):(1)$$S=\frac{m_{t}-m_{o}}{m_{o}}\cdot 100(\%)$$

Disintegration of the networks was checked after four weeks of swelling in PBS at temperature of 37 °C. The samples were dried and their weight was monitored until a constant value was reached (*m_dry_*). The mass loss of each hydrogel was determined as:(2)$$\mathrm{Hydrogel}\;\mathrm{mass}\;\mathrm{loss}=\frac{m_{o}-m_{dry}}{m_{o}}\cdot 100\;(\%)$$

### 2.7. Delivery

The drug-loaded hydrogel samples were prepared by adding 0.5% wt. neomycin to the polymer/protein/peptide solutions (composition given in Table 1) under continuous stirring. The amount of loaded drug was carefully chosen [14,46], taking into account the pharmaceutical prescriptions [46] as well as the adverse effects or local toxicity [47]. The hydrogel samples were prepared by using the same freezing/thawing method mentioned above (Section 2.2). In this way, the neomycin molecules were effectively entrapped in the network. Furthermore, a quantity of the absorbed drug could be efficiently distributed in the tissues, contributing to accelerated wound healing [10].

In vitro drug release characteristics were evaluated by the dialysis method. A sample of 0.2 g was introduced into a dialysis bag (the molecular weight cut-off being 10 kDa) immersed in a closed bottle with PBS solution (pH = 7.4) at a constant temperature of 37 °C in a thermostatic chamber. At different time intervals, aliquots of 1 mL were withdrawn from the release medium and replaced by 1 mL pre-heated PBS solution, in order to maintain a constant volume. The quantity of neomycin in the release medium was determined by UV spectrophotometry using a standard calibration curve (a linear dependence of absorbance versus drug concentration was obtained, R^2^ = 0.9924). The absorbance was recorded at 202 nm using a Libra UV-Vis spectrophotometer (Biochrom Libra S35PC, UK). The drug release studies were performed in triplicate, and the UV absorbance was continuously determined as the samples were collected in order to assure the drug’s stability in the investigated environment. The experimental data were analyzed using OriginPro 8.5 software to generate linear regression fits. The minimum values of the residual sum of the squares (*RSS*) were considered for identifying the best-fitting model [48,49]. However, the number of parameters (*p*) involved in a given model influences *RSS* values. The Akaike Information Criterion (AIC) [50] was used for a statistical analysis of the drug release mechanism, being independent of the number of parameters introduced by each model:AIC = *N* ln(*SSR*) + 2 *p*(3)
where *N* represents the number of experimental data.

According to this criterion, the model with the smallest AIC value was taken into consideration as the best-fitting model to describe the drug release mechanism [48,49].

## 3. Results

### 3.1. Morphology of the Hydrogels

After three cycles of freezing/thawing were applied to polymer/protein systems of different compositions (Table 1), hybrid networks were formed due to the multiple interactions that were established between the functional groups belonging to unlike or like macromolecules, and between GSH and BSA. The morphology of the hybrid networks was examined by SEM using lyophilized samples. The structure of hydrogels is porous, and the pores are interconnected (Figure 1). In crystalline aggregates, PVA-PVA junctions are created during a slow thawing [11,12,13,14] and they act as knots of a stable network. The polymer segments from the amorphous domains (PVA and PULL long chains) ensure the pore connectivity in the physical network. On the other hand, in the presence of cations from PBS, the monomeric units of PULL interact favorably with the hydrophilic groups of PVA [51,52].

The presence of a high number of amino acid residues (583) influences the BSA features, which allow interactions with drugs or bioactive compounds with broadly differing properties (anionic or cationic, hydrophobic or hydrophilic, etc.). The protein presents an oblate shape, consisting of domains I, II, and III (Figure 1), stabilized by disulfide bonds. Its structure incorporates 35 Cys residues forming 17 intramolecular disulfide connections and 1 free -SH group. The disulfide bridges located in relatively close proximity create short loops in the structure. Thus, pure BSA is composed of 67% α-helices and β-sheets, 10% β-turns, and 23% random structures [53]. BSA presents secondary and tertiary structures that are sensitive to pH, temperature, and denaturing agents. Circular dichroism spectroscopy and conductive atomic force microscopy investigations have shown that primary and secondary BSA structure changed after binding drugs [26]. By using spectroscopy and molecular docking, it has been shown that GSH is able to spontaneously bind via hydrogen bonds to BSA amino acid residues localized at the site I in the sub-domain IIA (Figure 1) through an enthalpy-driven process [54].

The pore size, determined as an average value of approx. 80 dimensions, ranges from 15.7 μm and 24.5 μm, depending on sample composition (Table 1). Generally, the low protein content does not significantly modify the pore dimensions of the PULL/PVA matrix. The increase in BSA concentration in the sample changes the hydrogel’s morphology.

Intermolecular interactions between -OH groups of polymers and the crystalline zones formed after applying successive freezing/thawing cycles act as network nodes. The amorphous domains consist of entangled polymer chains that create the connection between the pores in the physical network. A high content of BSA favors its association in clusters and aggregates in the amorphous region. This tendency disturbs the overall network structure which becomes weaker and the presence of rich protein domains determines a decrease of average pore size (samples 5 and 6) (Figure 1f,g,i). The hydrogel with 90% BSA exhibits a compact wall structure of pores with an average pore size of 24.5 μm (sample 7, Figure 1g).

In addition, for high BSA content, the protein can form large aggregates. As recently shown for another PVA-based system (PVA/HPC/BSA [19]), the hydrogels’ morphology can be tuned during their preparation by selecting the appropriate composition and carefully choosing the number of cycles during the freezing/thawing process.

In particular, sample 8 (containing 2% lysozyme) presented two types of pores: larger pores with an average size of 20.3 μm and smaller pores on the polymer matrix walls with an average size of 0.87 μm (Figure 1j). A possible explanation is the establishment of strong intermolecular bonds of lysozyme with the polymer matrix, influencing the structure and morphology of the network, with consequences for rheological behavior and drug delivery kinetics.

### 3.2. Infrared Spectroscopy of Hydrogels

#### 3.2.1. Interactions in the Polymer Matrix in Absence and Presence of Proteins

The specificity of the BSA—polymer matrix interactions were investigated by infrared spectroscopy in ATR geometry. In the absence of BSA, the infrared spectrum of the PVA/PULL control hydrogel is almost identical to that of native PVA [15], having, in addition, the ν(C-OH)+ν(C-C-O) band of PULL at 1035 cm^−1^. As presented in Figure 2a, the crystalline sensitive band ν(C-OH) of PVA was observed at 1144 cm^−1^, suggesting that the intermolecular H-bonding between the PVA and PULL took place in the disordered regions.

Instead, the characteristic ν(C-OH)+ν(C-C-O) band of PULL in the control hydrogel showed a depletion of the component at 993 cm^−1^, which is associated with the primary alcohol at the C6 position and it is a measure of the strength and extent of interchain interactions [55]. Disappearance of the 993 cm^−1^ band suggests the partial destruction of the intermolecular associated structure of PULL chains. As a result, the intermolecular interactions took place between the separated macromolecules from the amorphous areas of PULL and those of PVA. Likewise, the slight amplification of the band at 1016 cm^−1^, connected with the amorphous domains [55], and its shift to 1024 cm^−1^, indicated the existence of H bonds weaker in intensity than those found in the pure PULL.

**Hydrophilic interactions.** The conformational changes of PVA and PULL induced by the interaction with BSA can be extracted by analyzing the characteristic signals of the alcoholic groups and etheric moieties from pyranose rings in the 1200 cm^−1^–900 cm^−1^ region. The difference spectra [PULL/PVA/BSA] (samples 2–7) [PULL/PVA] (sample 1) from Figure 2b show the successive development of several types of intermolecular interactions. At low BSA concentrations (5–10%), a new network of H bonds was formed in which the primary alcohol groups of PULL were involved, suggested by the positive bands at 1017 cm^−1^ and 998 cm^−1^. The band at 1017 cm^−1^, assigned to H-bonded [ν(C6-O6H)•••)], grew and became very intense for sample 5 (50% BSA). Meanwhile, the intensity increase in ν_asym_(CH_2_) at 2918 cm^−1^ (shifted here from 2926 cm^−1^ from pure PULL), suggests the development of hydrophobic interactions. The negative band at 1096 cm^−1^ belongs to the hydroxyl groups of PVA from the amorphous domains that were redirected for BSA binding.

For samples 4 and 5, positive bands appear between 1040 cm^−1^ and 1055 cm^−1^, and they belong to PULL. It appears that a network of intermolecular H bonds with mainly BSA expanded with the help of secondary alcohols C3-O3H. It must be emphasized that BSA has many residues (according to PDB 3V03) situated mainly in loops (Ser 58, 109 and 272; Thr 438; Tyr 84; Asp 107, 108, 111, 295 and 363; Glu 95, 171, 293 and 299 etc.), representing less than 10% of total number per subunit, that could be actively involved in hydrogen bonds [45]. A further increase in BSA concentration leads to a decrease in the components at 1022 cm^−1^ (PULL) and at 1070 cm^−1^–1080 cm^−1^ (mostly PVA), indicating the strong mixing at the molecular level of the individual chains of PULL and PVA with BSA. The crystallinity band of PVA at 1144 cm^−1^ [ν(C-OH)] did not undergo major changes, even for high BSA concentrations. It can be concluded that the interactions with BSA are limited to the amorphous and interface areas of PVA and PULL.

In the region of the stretching vibrations corresponding to hydroxyl groups, the intense ν(OH) of PVA (3298.8 cm^−1^) had blue shifted to 3306.8 cm^−1^ in sample 1, and, later on, changed its shape and position when the concentration of BSA increased. For 50% BSA in the sample, ν(OH) changed, in appearance, to ν(NH). It was positioned at 3293 cm^−1^, as in pure BSA, thus suggesting that most of the protein molecules preserved their native conformation.

**Hydrophobic interactions.** Upon adding BSA, the red shift of ν_sym_(CH_2_) of PVA from 2915 cm^−1^ to 2912 cm^−1^ and the blue shift of ν_asym_(CH_2_) from 2936 cm^−1^ to 2938 cm^−1^ were visible up to 50% BSA, and showed the hydrophobic interactions of the PVA chain with both BSA and PULL macromolecules under the strong influence of BSA. Above 30% BSA, a new band appeared at 2918 cm^−1^, and that shifted to 2925 cm^−1^ at 70% BSA. This band resembled ν_asym_(CH_2_) of PULL at 2926 cm^−1^, but red-shifted by 8 cm^−1^ at 30% BSA. This fact suggests that for BSA content between 30% and 70% BSA, the hydrophobic interactions between PULL and BSA become much more intense than they were at lower concentrations of BSA. In this sense, it must be emphasized that 61 Leucine and 46 Alanine residues were found in a single BSA monomer, and their side chains could be involved in hydrophobic interaction. Above 70% BSA in hydrogel composition, the hydrophobic interactions of BSA were very evident by the red shift of ν_asym_(CH_2_) from 2936 cm^−1^ in simple BSA to 2934 cm^−1^ at 90% BSA, and the blue shift of ν_sym_(CH_3_) from 1391 cm^−1^ to 1397 cm^−1^.

#### 3.2.2. Conformational Changes of BSA in Presence of PULL/PVA Mixtures

Among the most important absorption bands specific to BSA, namely Amide A, Amide I, and Amide II, only the Amide A band (3292 cm^−1^, ν(NH)] maintained its profile and position at 90% and 70% BSA in the PULL/PVA mixture (Figure 2a). The Amide I (1647 cm^−1^) and Amide II bands (1522 cm^−1^) were an envelope of the vibrations describing the secondary structure elements, the amino acids from the side groups, and the C=O and NH vibrations of the polypeptide chain. In the PVA/PULL/BSA samples, Amide I and Amide II were narrower, and were blue shifted by 10 cm^−1^ at 10% – 30% BSA and by 5 cm^−1^ for 50%–90% BSA.

The structural changes of BSA in the PVA/PULL hydrogels can be extracted from a combined analysis involving the difference spectra [PULL/PVA/BSA]—[BSA] illustrated in Figure 2c and the quantitative analysis of the BSA secondary structure. In this regard, the Fourier self-deconvolution, the second derivative, and the curve-fitting by Voigt functions of the Amide I band allowed us to separate the main components and their positions, as follows: 1685 cm^−1^ (“free” carbonyls residing in hydrophobic regions and β-turns), 1670 cm^−1^ (turns and “free” carbonyl groups in a polar environment), 1653 cm^−1^/1648 cm^−1^/1640 cm^−1^ (long and short α-helices, and 3_10_-helices, that can be arranged in regular structures and tight bundles), 1630 cm^−1^ and 1625 cm^−1^ (short-chain segments connecting two consecutive α-helices), and 1604 cm^−1^ (scissoring of -NH_2_ groups and extended chains) [56,57,58]. The variations in the secondary structure elements of BSA are presented as a histogram in Figure 3a, and are expressed as intensity variations of the component bands in Figure 3b.

Thus, the content of the regular structures of BSA decreases from approx. 75% in the initial state to approx. 72% for sample 2 (5% BSA), and down to 66–64% for sample 3 (10% BSA). The negative band in the difference spectra [PULL/PVA/BSA]—[BSA] centered on 1690 cm^−1^ (Figure 2c) correlates with the negative values of ΔA of the free carbonyl groups from the hydrophobic areas of BSA (from loops and turns) (Figure 3b). The negative intensity variations of the band at 1630 cm^−1^ are correlated to the vibrations of the “bound” carbonyl groups belonging to the residues in the loop areas connecting two α-helix structures. These chain segments are very flexible, and the above observations suggest that they interact with the polymer matrix more strongly and quickly than the rigid helix domains. The unfolding of the α-helix involves about 7% of the total protein (over the entire range of BSA content used in the mixture with polymers), and it appears up to about 30% BSA. Structures of the short α-helices (1648 cm^−1^) are more affected (≈ 23% for 10% BSA) compared to long and compact α-helices (1653 cm^−1^) (the maximum is 4%, observed for 50% BSA). This would also explain the increase in hydrophobic interactions observed both in the histogram of secondary structures of BSA (Figure 3a) and from the intensity variations (associated with structural changes) in Figure 3b. This statement can be correlated with the observation from the above section regarding the analysis of hydrophobic interactions, in the area of ν(CH_2_) (2800 cm^−1^ –3000 cm^−1^). These hydrophobic interactions were stronger for BSA content between 30% and 70% in the polymer matrix. This conclusion results from the three types of analyses.

Therefore, the mixture of BSA with PVA/PULL did not seriously impact the secondary structure of the protein. The long and compact α-helices are interacting, most probably, at the level of the side–side interactions with the matrix, and the short helices and segments were fully engaged in intermolecular interactions with PVA/PULL.

For sample 8, the FTIR spectrum was similar to sample 1 and also contained the Amide I and Amide II from lysozyme. Amide I shifted to 1652 cm^−1^ (from 1641 cm^−1^ in lysozyme) and Amide II was found at 1547 cm^−1^, significantly reduced in amplitude, and shifted from 1532 cm^−1^ in the original lysozyme (Figure 4). The ν(OH) of PVA/PULL was observed at 3291 cm^−1^, with a downshift of 24 cm^−1^ as compared to the control sample.

The position of the Amide I maximum coming from the lysozyme indicates the stabilization of the α-helix structures in the mixture and the involvement of the disordered regions in the intermolecular bonds with the polymer matrix. The GSH characteristic signals are overlapped by the more intense vibrations of the matrix.

### 3.3. Rheological Behavior

Oscillatory and continuous shear measurements revealed a decrease in value of the rheological parameters of PULL/PVA hydrogels in the presence of protein (Figure 5, Figure 6 and Figure 7). An exception was observed for low BSA content (5%), when the –OH groups of polymer (PVA or PULL) from the amorphous zone interacted with amide groups from the short unfolded helices and segments of BSA and improved the network strength (as shown by G’ and σ_y_ values). Higher BSA content (w_BSA_ ≥ 10%) contributed to the formation of a weaker network structure; a diminution of the G′, G” and η values was observed. However, the presence of 2% lysozyme in the PVA/PULL matrix determined a slight decrease in G’, while the σ_y_ value increased from 63 Pa to 79 Pa, possibly due to the heterogeneous structure of sample 8 (a tendency of phase separation was observed in solution state for higher protein content).

In the presence of BSA, the rheological parameters of the PULL/PVA matrix registered lower values. In frequency sweep experiments, the viscoelastic moduli, G′ and G″, showed a slight variation with the oscillation frequency (ω). The loss tangent values were between 0.08 (samples 1 and 2) and 0.1 (samples 3–8), suggesting that a stable network structure was formed in all cases. The physical strength of the hydrogel (indicated by G′ and σ_y_ values) slightly decreased with the increased BSA content (samples 2 to 7).

At low shear rates, the Newtonian viscosity was obtained (Figure 6), and its values are given in Table 1. With the increasing shear rate, above 0.03 s^−1^, the viscosity began to decrease and scaled as ${\dot{\gamma }}^{-a}$, where *a* takes values between 0.68 and 0.77 as the BSA content decreases.

Yield stress (σ_o_) represents the minimum value of shear stress necessary for initiating the shear flow. This parameter was determined for each sample in shear flow conditions, following the shear viscosity variation with the applied shear stress (Figure 7, Table 1). An analysis of the experimental values of the yield stress values obtained for hybrid hydrogels revealed that the sample composition influences the network strength; high polymer content determines high σ_o_ values. In all rheological tests, a synergistic behavior was observed for sample 2 with low content of BSA (5%), for which σ_o_ = 170 Pa.

### 3.4. Swelling

The swelling investigations were carried out in PBS solution. Figure 8 illustrates the swelling curves for the hydrogel samples at pH = 7.4. For high content of BSA (50% or 70% BSA), a faster initial swelling rate was noticed after the hydrogel’s immersion, and then the swelling rate slowed down. Samples 2, 3, 4, and 8 presented a continuous increase in the swelling degree and no burst effect was registered. The maximum values of swelling at equilibrium (S_eq_) depended on the hydrogels composition and their stability in the PBS environment (Table 1).

At high BSA concentrations, the swelling degree of the hydrogels increased, but, at the same time, the loss of mass rose. This is a result of the disintegration process that occurs in time during the networks immersion in PBS (Table 1). The samples with high BSA content (70% and 90%) presented aggregates of protein trapped within the polymer matrix (Figure 1f,g). These aggregates harmed the hydrogels’ integrity, leading to the sample’s disintegration in PBS environment. After four weeks of immersion in the buffer solution, a considerable mass loss was observed (36.14% for sample 6 and 59.77% for sample 7).

In order to discuss the nature of the solvent diffusion through the hydrogel pores, the following equation was used:(4)$$F(t)=\frac{m_{t}}{m_{\infty }}=k_{s}\;t^{n_{s}}$$
where F(t) is the total water uptake at a given time *t*; $m_{t}$ and $m_{\infty }$ are the amount of solvent absorbed by the network at a given time, *t*, and when the swelling equilibrium is attained, respectively; $k_{s}\;$ is a characteristic rate constant influenced by the network structure; and $n_{s}$ is a transport number indicating which phenomena control the swelling process: diffusion and/or relaxation.

Equation (4), applied to the early swelling stage, gave linear dependences of the function F(t)
*versus* time (*t*). From the slope of these dependences, the diffusion exponent, $n_{s}$, was obtained (Table 1). The values of this exponent reflect the solvent transport mode through the hydrogel pores. For all samples, $n_{s}$< 0.5, suggesting that the diffusion is quasi-Fickian.

### 3.5. Neomicin Delivery

The release of a drug from a hydrogel in the surrounding fluid is regarded as a mass transfer from a high drug concentration (dosage form) to a domain with low drug content. The drug molecules can either diffuse among the swollen hydrogel or can be adsorbed by the system components, their motion being slowed down. The drugs’ movement is influenced by the drug–network interactions, temperature, pH etc. Different mathematical models have been developed over time, and they are used to predict the drug release process through a series of parameters which are useful for achieving an optimum formulation [59].

In the present study, the delivery of neomycin incorporated into the hydrogels was assessed by using the following equations:

Korsmeyer–Peppas [60]:*M_t_/M_∞_ = k t^n^*(5)

First-order [59]:*ln(1 − M_t_/M_∞_) = −k_11_ t*(6)

Higuchi [61]:*M_t_/M_∞_ = k_h_ t^1/2^*(7)

Peppas–Sahlin [62]:*M_t_/M_∞_ = k_1_ t^m^ + k_2_ t^2m^*(8)
where *M_t_/M_∞_* represents the fraction of released drug at the time *t*. Generally, *k* denotes the release kinetic constant according to different models. The constant *k* from Equation (5) is correlated with the characteristics of the administered drug [40]. *k_11_* and *k_h_* are the first-order and Higuchi release rate constants, respectively. *k_1_* and *k_2_*, from Equation (8), are two kinetic constants correlated with the Fickian or non-Fickian contributions to diffusion, respectively, in the global process of drug delivery [62]. Equation (5) is valid until the drug release attains 60% of its maximum level. The value of release exponent, *n*, may suggest the drug release mechanism from a polymeric network. Thus, for cylinder-like geometry, *n* = 0.89 and it corresponds to zero order (time independent) release or case II transport; *n* > 1 Super Case II transport; 0.45 < *n* < 0.89—anomalous transport phenomena controlled by drug diffusion and network swelling; *n* = 0.45 indicates a Fickian diffusion; *n* < 0.45—pseudo-Fickian diffusion with slow drug release [60,63,64].

The exponent *m* from Equation (8) characterizes the pure Fickian diffusion for materials presenting controlled drug delivery [59,62].

Figure 9 shows the profiles of neomycin release from six selected hydrogel samples in PBS at 37 °C (pH = 7.4, at which fewer amino groups of drug molecules are protonated). For the hydrogels with a high density of intra- and intermolecular interactions, the release of neomycin may be delayed or even strongly diminished after 480 min (as was observed for sample 1, the highest drug amount is released in the first 60 min). The presence of proteins considerably improves the neomycin delivery, and the most successful formulations result to be samples 3, 4 and 8. For 30% BSA, it was observed from the FTIR spectra that the hydrophobic interactions were relatively weak, allowing for the progressive release of the drug. It was found experimentally that 2% lysozyme is an optimal concentration that allows the formation of stable hydrogels with PULL/PVA matrix (sample 8), whereas a higher content of protein introduced into the polymer matrix causes a phase separation. The efficiency of this hydrogel is also due to the presence of pores of different sizes, as shown in Figure 1h,j.

Table 2 presents the kinetics results obtained by fitting the experimental data according to four selected releasing models. For all investigated samples, the Korsmeyer–Peppas model (Equation (5)) is adequate to describe the in vitro drug release mechanism for neomycin (low AIC values). The Peppas–Sahlin approach is more suitable for hybrid hydrogels (samples 2–8). For pure PVA hydrogels, the transport mechanism was non-Fickian diffusion leading to zero-order release behavior as the number of freezing/thawing cycles increased [65]. According to recent investigations [19,25], three freezing/thawing cycles were selected in the present study, ensuring the formation of the interpenetrated porous networks for hybrid hydrogels (Figure 1). The incorporation of the drug into the hydrogel was performed by dissolution of the drug into the polymer/protein solution before submitting it to freezing/thawing cycles and then freeze-drying by lyophilization. Thus, the entire amount of the drug was retained in the hydrogel. During the first 30 min, 0.2 < *n* < 0.32 (Table 2), suggesting that the pseudo-Fickian diffusion characterized the neomycin release. If the relaxation has no influence on the release mechanism, the values of *n* (Equation (5)) and *m* (Equation (8)) would be very close. For the investigated hydrogel sample, there was a difference between the *n* and *m* exponents (Table 2) suggesting that the pseudo-Fickian diffusion was accompanied by chain relaxation. Differences in the release behavior of the investigated samples appeared after 30 min, and hybrid hydrogels (samples 4, 5, and 8) presented a more sustained release (Figure 9). After 8 h, the release profiles for samples 3, 4 and 8 were very close. Sample 5 presented a faster rate of neomycin release during the first 4 h, but it slowly decreased after this period of time.

The mechanism of drug release is complex, being controlled by swelling, diffusion, chain relaxation, or erosion of the matrix, and it is dependent on the interaction between drug and macromolecules from the hydrogel. The diffusion process is governed by possible electrostatic interactions (fewer amino groups of neomycin are protonated at pH = 7.4, when BSA is negatively charged; phosphate buffer presents a ionic strength of 50 mM, and the electrostatic charges of protein could be partially shielded in presence of competing ions), or by steric interactions with the network and, for high BSA content (≥70%), by bulk degradation of the hydrogel in the release fluid at 37 °C. Since the investigated hydrogel samples did not completely degrade during the release experiment, the drug was unable to be released completely (Figure 9). As a result, some of the drug molecules that were bound via intermolecular interactions with the hydrogel macromolecules remained entrapped in the matrix. This effect diminished in the presence of proteins (less that 50% compared with the polymer matrix) when the delivery was considerably improved, demonstrating a synergistic effect of all hydrogel components. Used as wound dressings, these porous hydrogels can provide a suitable wet environment and, by absorbing wound exudates, they reduce the risk of infection. It was shown that the pH values of chronic wounds are between 7.15 and 8.9, and they slowly decrease during healing [10,66]. Antibiotics play an important role; their presence accelerates the wound curing and scar formation [33]. Among the topical antibiotics, neomycin is an effective drug with bactericidal and bacteriostatic action for the skin and mucous membrane infections [67].

Many efforts were undertaken to select the most appropriate method for quantitative determination of neomycin [68,69]. A sensitive detection and good reproducibility were obtained by liquid chromatography with pulsed electrochemical detection [68] or evaporative light scattering detection [69]. A reverse phase liquid chromatographic method with a charged aerosol detection, coupled with mass spectroscopy, was used by Stypulkowska et al. [40] for a quantitative determination of neomycin sulfate in pharmaceutical preparations (accuracy > 95.8%). A fluorescent sensor based on quantum dots with two selective binding sites able to detect neomycin in biological systems was reported by Wan et al. [41]. In order to maintain the drug’s concentration during its release, injectable hydrogels were designed by using chitosan, alginate, and PVA, crosslinked with tetraethoxysilane [70]. The amount of neomycin released from hydrogels was also determined by UV visible spectrophotometry, and the effectiveness of this drug was comparable with that of other antibiotics.

Thus, the sensitivity of the UV detection of neomycin is comparable with the aforementioned methods. In our study, the possible errors were diminished by measuring the UV absorbance as the samples were collected. Similar results have been reported in the literature for the neomycin delivery [70]. According to previous investigations, the stability of neomycin was not affected for two or three weeks when its solution was stored in a refrigerator [37,71], or loaded in 3D printed poly-L-lactide mats and delivered in vitro at 37 °C [72]. An in vitro drug release study on ectosomes loaded neomycin showed 65–80% delivery after 8 h and good stability (no aggregation or decrease in drug content) after 45 days [46]. Therefore, the experimental conditions during the hydrogel preparation and release studies did not affect the structural integrity of the incorporated drug or protein.

The presence of neomycin in various materials slows down the growth of Gram-negative or Gram-positive bacteria [36,37,38,39,40,41,70]. In addition, we expect an improvement of biological properties using GSH and lysozyme, this aspect being of interest for a future study.

## 4. Conclusions

Natural and synthetic polymers were used to prepare hydrogels in the presence of two proteins, BSA and lysozyme, by using nontoxic approaches. The sample composition and the preparation steps were carefully selected in order to achieve network structures which are able to incorporate drug molecules and release them in simulated physiological fluid (phosphate buffer solution at pH = 7.4 and temperature of 37 °C). The networks presented elasticity and yield stress, conferring stability and memory effects when submitted to viscoelastic stress. For PULL/PVA/BSA hydrogels, SEM analysis revealed network structures with interconnected pores with sizes between 15.7 µm and 24.5 µm. The sample containing 2% lysozyme displayed both large and small pores, with average sizes of 20.3 μm and 0.87 μm, respectively. Infrared spectroscopy analysis confirmed the interplay of intermolecular hydrogen bonds and hydrophobic interactions (especially between 30% and 70% BSA content).

The swelling kinetics is characterized by a non-Fickian diffusion of solvent molecules through the hydrogels’ pores. The release of an active substance (neomycin, an antibiotic with bactericidal and bacteriostatic action), incorporated before the gelation process, is governed by the pore size of the hydrogel matrix and the strength of established interactions. Two models, developed by Peppas and coworkers, provided the best fitting results of the experimental delivery data. Thus, these hybrid hydrogels are suitable for wound dressing applications, and they could be effective carriers for the delivery of antibiotics.

## Data Availability

Data are available upon request.

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
