# Peer review of "Hybrid Hydrogels for Neomycin Delivery: Synergistic Effects of Natural/Synthetic Polymers and Proteins"

_polymers, 2023, doi:10.3390/polym15030630_

Round 1

Reviewer 1 Report

Comments:

1.     Why Figure 1i and j has smaller size compared to other figures? Please make them as the same size.

2.     In line 299, please remove the red highlight.

3.     What is the biocompatibility of this hydrogel, I would recommend the author add some cell cytotoxicity experiment since biocompatibility is important for the hydrogel in drug delivery area.

4.     What is the loading amount of BSA and the neomycin sulfate in the hydrogel sample?

5.     In Figure 8, why there is no sample 7 data here? Similarly, in Figure 9, why there is no sample 6 and 7 data here?

6.     I would also recommend the author add some biological tests of the released neomycin sulfate, since no efficacy studies data here, we won’t know if the therapeutic effects will be influenced by the hydrogel.

Author Response

  1. Why Figure 1i and j has smaller size compared to other figures? Please make them as the same size.

We corrected, all figures have the similar size.

  1. In line 299, please remove the red highlight.

Red highlight was removed.

  1. What is the biocompatibility of this hydrogel, I would recommend the author add some cell cytotoxicity experiment since biocompatibility is important for the hydrogel in drug delivery area.

At this stage, the main goal of the paper was to report new hydrogels and their main characteristics. For the time being, we did our best to improve the revised manuscript. The suggested cytotoxic assays will be taken into account for an additional study which will be accomplished to achieve results in terms of biocompatibility and cytotoxicity.

  1. What is the loading amount of BSA and the neomycin sulfate in the hydrogel sample?

The BSA concentration in the sample is given in Table 1 (column 2).

The loading amount of Neomycin is 0.5% wt. (Section 2.7).

  1. In Figure 8, why there is no sample 7 data here? Similarly, in Figure 9, why there is no sample 6 and 7 data here?

Swelling data for sample 7 were included in Figure 8. The non-homogeneity of the samples with high protein content (BSA agglomerates) has an impact on diffusion. In addition, the hydrogel mass loss was significantly higher and the corresponding errors too, thus the data for sample 6 and 7 were not included in Figure 9.

We suppose that some neomycin molecules are „arrested” in the aggregates formed by BSA in the amorphous region of the hydrogel.

  1. I would also recommend the author add some biological tests of the released neomycin sulfate, since no efficacy studies data here, we won’t know if the therapeutic effects will be influenced by the hydrogel.

We thank the reviever for the careful analysis of the manuscript.

We will take into consideration the reviewer’s recommendation and a series of experiments will be planned to evaluate the biological properties of the hybrid hydogels.

According to literature data, the presence of neomycin in various materials slows down the growth of Gram-negative or Gram-positive bacteria. In addition, we expect an im-provement of biological properties using GSH and lysozyme, this aspect being of interest for a future study.

We thank the reviewer for the careful analysis of the manuscript and useful comments and suggestions.

Reviewer 2 Report

Hybrid Hydrogels for Neomycin Delivery: Synergistic Effects 2 of Natural/Synthetic Polymers and Proteins 

Comments:

1. Introduction: the more review on role of GSH and lysozyme on wound dressing is needed with previous research works and supporting references. Additionally, the mechanistic on freeze-thaw cycling on intermolecular interaction should be described in Introduction for profoundly understanding with supporting related works.  

2. Method: the component of sample formula should be addressed separately in detail for clearly know each compound in % of formulation. Please be informed program type for release profile fitting to mathematic model.  Practically, the analysis for neomycin has been approved for HPLC or microbial assay, not UV-Vis spectrophotometer owing to its complicate structure and instability. The release tests were done without using any membrane. The formulations were dispersed in media and the release from other components might interfere the analysis. Statistical analysis should be done, and significance added to text and graphs.

3. Results: The viscosity values should be presented. The sample size of each investigation should be added in each figure or table again. The more discussion is needed with sufficient supporting references for 3.2.1-3.2.3 and 3.4, 3.5. 

4. The drug content should be checked before drug release. And the duplicate test seemed not enough for drug release. Please explain the incomplete drug liberation.

5. The more explanation based on the author result of release behavior as pseudo-Fickian diffusion should be conducted. The explanation for obtained estimate parameters from Peppas-Sahlin equation should be undertaken and related to the release results and related works.

6. The bioactivities relied on function of GSH and lysozyme have not seen clearly.

7. For neomycin loading, so please exhibit the antimicrobial activities of developed devices. In addition, the discussion on drug analysis method and stability should be addressed in results. 

Author Response

  1. Introduction: the more review on role of GSH and lysozyme on wound dressing is needed with previous research works and supporting references. Additionally, the mechanistic on freeze-thaw cycling on intermolecular interaction should be described in Introduction for profoundly understanding with supporting related works.

We thank the reviever for the  careful analysis of the manuscript. The introduction was reviewed and new aspects were introduced concerning GSH and lysozyme, as well as about the PVA hydrogels formed by freezing/thawing method.

  1. Method: the component of sample formula should be addressed separately in detail for clearly know each compound in % of formulation. Please be informed program type for release profile fitting to mathematic model.  Practically, the analysis for neomycin has been approved for HPLC or microbial assay, not UV-Vis spectrophotometer owing to its complicate structure and instability. The release tests were done without using any membrane. The formulations were dispersed in media and the release from other components might interfere the analysis. Statistical analysis should be done, and significance added to text and graphs.

We did our best to improve the manuscript. During the manuscript revision, we repeated the release experiments.

Thus, the drug release studies were performed in triplicate, the UV absorbance was continuously determined as the samples were collected, in order to assure the drug stability in the investigated environment. The experimental data were analyzed using OriginPro 8.5. to generate linear regression fits.

According to literature data, the sensitivity of the UV detection of neomycin is comparable with chromatographic methods and, in our study, the possible errors were diminished by measuring the UV absorbance as the samples were collected.

Moreover, UV Spectrophotometry is a well-established and convenient technique, widely used in laboratories for such studies due to the fact that is easy and fast to use, is accessible for laboratory research, and does not require a large amount of solvent. In contrast, the HPLC separation method is time-consuming, the organic mobile phase quite expensive and the number of colected samples was huge and difficult to be processed in reasonable time. However, the most promising delivery systems will be selected in the future and HPLC with UV detection will be used for deeper investigations.

It was observed by other authors that the stability of neomycin is not affected during two or three weeks when its solution is stored in a refrigerator [Ayliffe, G.A.J. Stability of neomycin resistance in Staphylococcus aureus. J. Clin. Path. 1970, 23(1), 19–23. https://www.uspharmacist.com/article/neomycin-sulfate-25-mgml-oral-solution. ]

The release tests were done using a dialysis bag with the molecular weight-cut off of 10 kDa. We tested firstly the system free of drug in similar conditions to correctly assess the amount of neomycin.

  1. Results: The viscosity values should be presented. The sample size of each investigation should be added in each figure or table again. The more discussion is needed with sufficient supporting references for 3.2.1-3.2.3 and 3.4, 3.5. 

For drug delivery studies, samples of 0.2 g dry hydrogel (along with the same content of PVA/PULL, BSA and GSH as given in Table 1) were prepared using the same procedure as for samples free of drug; they were loaded with 0.5% neomycin sulfate before being submited to freezing/thawing cycles and then they were freeze-died by lyophilization. The comments have been improved in all sections.

  1. The drug content should be checked before drug release. And the duplicate test seemed not enough for drug release. Please explain the incomplete drug liberation.

The incorporation of drug in the hydrogel was performed by solubilization of the drug in the polymeric solution before being submited to freezing/thawing cycles and then they were freeze-died by lyophilization. Thus, the entire drug amount was retained into the hydrogel.

The drug delivery is controlled by diffusion process, by possible electrostatic interactions (less amino groups of neomycin are protonated at pH = 7.4 when BSA is negatively charged; phosphate buffer presents a ionic strength of 50 mM and the electrostatic charges of protein could be partially shielded in presence of competing ions), but also by steric interactions with the network and, for high BSA content (³ 70%), by bulk degradation of the hydrogel in the release fluid at 37 °C. Since the investigated hydrogel samples did not completely degrade during the release experiment, the drug was unable to be released completely (Figure 9). As a result, a part of the drug molecules that were bound via intermolecular interactions with the hydrogel have remained entrapped in the matrix.

Generally, the release of the drug is a complex phenomenon, controlled by matrix swelling, diffusion through the network, osmosis, and also surface or bulk degradation of the hydrogel.

Since the hydrogel did not completely degrade during the release experiment, the drug was unable to be released completely. As a result, a part of the drug that was bound via inter- and intramolecular interactions with the hydrogel molecules remained entrapped in the matrix.

The UV absorbance was continuously determined as the samples were collected, in order to assure the drug stability in the investigated environment.

In the revised manuscript, the drug release experiments were performed in triplicate.

The experimental data were analyzed using OriginPro 8.5. to generate linear regression fits.

  1. The more explanation based on the author result of release behavior as pseudo-Fickian diffusion should be conducted. The explanation for obtained estimate parameters from Peppas-Sahlin equation should be undertaken and related to the release results and related works.

We repeated the release experiments and in the revised manuscript we improved the comment at this section.

  1. The bioactivities relied on function of GSH and lysozyme have not seen clearly.
  2. For neomycin loading, so please exhibit the antimicrobial activities of developed devices. In addition, the discussion on drug analysis method and stability should be addressed in results.

At this stage, the main goal of the paper was to report new hydrogels and their main characteristics. For the time being, we did our best to improve the revised manuscript.

The suggested analyses will be taken into account for an additional study which will be accomplished to achieve results in terms of biocompatibility and cytotoxicity.

A discussion concerning the used drug analysis method compared with those presented in literature was included in the revised manuscript.

We thank the reviewer for the careful analysis of the manuscript and constructive comments and suggestions.

Round 2

Reviewer 1 Report

This work has been updated from previous reviews and I support publication of the work

Reviewer 2 Report

The authors improve the manuscript with additional review literature and new experiment and profound discussion that suite for publication in Polymers. The response to comment is comprehensive and valuable sufficiently.